# Copper connections: coordinating transport, sensing and systemic signalling in plants

Ju-Chen Chia, Tetiana-Olena Zavodna, Hanna Shatokhina, Yana Kavulych and Olena K. Vatamaniuk

Plant Biology Section, School of Integrative Plant Sciences, Cornell University, USA

copper homeostasis; copper transport; copper chaperons; copper sensing; copper signaling.

**Corresponding author:**
Olena K. Vatamaniuk;
Email: okv2@cornell.edu

**Associate Editor:**
Prof. Ingo Dreyer

## Abstract

Copper is an essential micronutrient that plays critical roles in plant metabolism, development and stress responses through its unique redox properties. While tightly regulated to prevent toxicity, labile copper also functions as a dynamic signalling molecule mediating developmental and environmental cues. Copper bioavailability in soils is influenced by complex physico-chemical factors, posing challenges for plant acquisition and homeostasis. Plants have evolved sophisticated mechanisms to regulate copper uptake, long-distance transport, intracellular trafficking and storage, balancing its essentiality with potential toxicity. This review summarizes current knowledge on copper homeostasis in plants, discusses uptake strategies in dicots and non-grass monocots, the coordination of internal copper transport and tissue distribution, and the emerging evidence for systemic copper signalling. Understanding these processes is important for improving crop nutrient use efficiency and resilience in mineral-deficient soils.

## 1. Introduction

Copper (Cu) is an essential mineral nutrient that readily cycles between the $Cu^{2+}$ and $Cu^{+}$ oxidation states. This redox flexibility allows Cu to serve as a cofactor in key metabolic enzymes involved in electron transfer reactions and essential biological processes such as photosynthesis, respiration and oxidative stress scavenging (reviewed in Broadley et al., 2012; Burkhead et al., 2009; Ravet & Pilon, 2013). In plants, Cu also plays roles in cell wall lignification, pathogen resistance and reproduction (reviewed in Rahmati Ishka et al., 2022). Beyond its static role as an enzyme cofactor, labile (loosely bound or bioavailable) Cu acts as a dynamic signalling metal and metalloallosteric regulator, increasingly recognized in cell proliferation and death pathways in animal systems (Chang, 2015; Chen et al., 2022; Ge et al., 2022; Wang et al., 2025). While the signalling functions of labile Cu in plants are less explored, $Cu^{+}$, the predominant form in the reducing environment of the cytosol, is essential for ethylene and salicylic acid perception and signalling, highlighting Cu as a dynamic and compartmentalized signal involved in developmental transitions and environmental responses (Azhar et al., 2023; Rodríguez et al., 1999; Schott-Verdugo et al., 2019; Wu et al., 2012). Recent studies also suggest that Cu can act as a systemic signal or influence an unidentified systemic signal in long-distance shoot-to-root communication of plant Cu status under varying environmental and developmental contexts (Chia et al., 2023).

Cu redox activity, however, poses toxicity risks, as its ability to accept and donate electrons can generate adventitious electron transfers to oxygen, causing oxidative stress (Ravet & Pilon, 2013). Moreover, $Cu^{2+}$ forms exceptionally stable complexes with organic ligands, as explained by the Irving–Williams series ($Ca^{2+}/Mg^{2+} < Mn^{2+} < Fe^{2+} < Co^{2+} < Ni^{2+} < Cu^{2+} > Zn^{2+}$), which ranks the relative stability of divalent metal-ligand complexes and highlights $Cu^{2+}$ as the strongest binder among biologically relevant metals; this high $Cu^{2+}$-ligand stability can lead to mismetalation and protein inactivation (Festa & Thiele, 2011; Irving & Williams, 1953; Martin, 1987; Osman & Robinson, 2023). Consequently, although total intracellular Cu may be in the micromolar range, the vast majority exists tightly bound to proteins and various cellular ligands, with the labile Cu pool estimated to be in the femto- to zeptomolar range (Ackerman et al., 2017; Hong-Hermesdorf et al., 2014). Intracellular Cu trafficking is mediated by chaperone proteins that ensure accurate metalation *via* protein-protein interactions and ligand exchange,

preventing free cytoplasmic Cu ion exposure during transit (Harrison et al., 1999; Meiser et al., 2011; Rahmati Ishka et al., 2022; Riaz & Guerinot, 2021; Wairich et al., 2022).

Historically, Cu was not among the earliest bioavailable metals; it became biologically indispensable with the rise of photosynthetic organisms and the oxygenation of Earth's atmosphere over 2.33 billion years ago (Bekker et al., 2004; Burkhead et al., 2009; Luo et al., 2016; Poulton et al., 2021). As atmospheric oxygen increased, iron (Fe), once dominant in early biochemistry, became less soluble due to $Fe^{3+}$ oxide formation, whereas $Cu^{2+}$ became more bioavailable, liberated from insoluble $Cu^+$ sulphide salts in oxygenated soils and waters (Burkhead et al., 2009). This environmental transition likely drove the evolution of Cu-dependent metalloenzymes and laid the foundation for the increasingly recognized, although not fully explored, intricate crosstalk between Cu and Fe homeostasis and their metabolic networks. The hallmark of this crosstalk is the increased uptake of Fe under Cu deficiency and *vice versa*, reciprocal regulation of some of Cu-responsive genes under Cu or Fe deficiency, modulation of Cu homeostasis through a Cu economy/metal switch component, and finding that Cu and Fe can partially substitute each other in long-distance signalling (Cai et al., 2024; Chia et al., 2023; Chia & Vatamaniuk, 2024; Kastoori Ramamurthy et al., 2018; Pätsikkä et al., 2002; Waters et al., 2012; 2014; Waters & Armbrust, 2013; Zhu et al., 2023). A detailed analysis of Cu–Fe crosstalk is beyond the scope of this review and is available in (Wairich et al., 2022; 2025).

Total soil Cu levels typically range from 2 to 100 mg/kg, but its bioavailability is controlled by complex physicochemical interactions, so only a small fraction is accessible for plant uptake (Alloway, 2013; Kabata-Pendias, 2000; McBride, 1994; Shorrocks & Alloway, 1988). Soil pH is a major determinant, with acidic soils increasing Cu solubility and alkaline soils promoting insoluble Cu compound formation, thereby reducing bioavailability (Alloway, 2013; McBride, 1994). Considering the tightness of Cu binding to organic matter, Cu bioavailability in organic soils is limited (Irving & Williams, 1953; McBride, 1994; Mitra, 2015; Shorrocks & Alloway, 1988). Clay minerals and soils with high cation exchange capacity adsorb Cu, further immobilizing it and limiting uptake (McBride, 1994). Additionally, redox conditions play a role in Cu bioavailability in soils: under waterlogged, anaerobic conditions, Cu may precipitate as Cu-sulphide (CuS), further decreasing its availability (McBride, 1994). These factors, along with agricultural practices such as the use of Cu-based fungicides and fertilization regimes, contribute to the dynamic behaviour of Cu in the soil-plant continuum. This review focuses on how plants harness Cu biochemical properties by utilizing a sophisticated Cu homeostatic system that regulates uptake, cellular trafficking, tissue distribution, storage and utilization. We also explore the differences in Cu uptake strategies between grasses and non-grass species, the proposed mechanisms of Cu sensing and emerging evidence for systemic Cu status signalling.

## 2. Overview of Cu homeostasis in plants

### 2.1. Cu uptake and long-distance transport in dicots and non-grass monocots

Copper predominantly exists as Cu(II) in aerated soils, where it forms complexes with organic matter, Fe and aluminium (Al) oxides. Thus, Cu must be mobilised before root uptake (Flemming & Trevors, 1989). In *Arabidopsis thaliana*, a model dicot, Cu(II)

is proposed to be reduced to Cu(I) before uptake, which is a mechanism likely shared by other dicots and non-grass monocots (Burkhead et al., 2009) and (Figure 1). This reduction is mediated by FRO4 and FRO5, members of the ferric reductase oxidase (FRO) family of membrane-bound enzymes. Following reduction, Cu(I) is transported into root epidermal cells mainly *via* the copper transporter (COPT) family. AtCOPT1 and AtCOPT2 facilitate high-affinity Cu(I) uptake (Bernal et al., 2012; Gayomba et al., 2013; Jung et al., 2012; Kampfenkel et al., 1995; Sancenon et al., 2003). In addition, AtZIP2, a member of the ZIP family better known for Zn and Fe transport, is transcriptionally upregulated under Cu deficiency (Bernal et al., 2012; Wintz et al., 2003; Yamasaki et al., 2009; Yan et al., 2017) and (Figure 1). Recent studies in *A. thaliana* revealed that ZIP2 localizes primarily to root epidermal cells, with additional signal detected in the vasculature of transgenic plants expressing a *ZIP2* genomic fragment fused to *mCitrine*. *zip2* mutants are hypersensitive to Cu deficiency and accumulate less Cu in both roots and shoots, supporting a role for ZIP2 in Cu acquisition at the root periphery and possibly in Cu partitioning between roots and shoots (Robe et al., 2025).

Long-distance Cu transport involves multiple steps: efflux from root xylem parenchyma cells into the xylem apoplast, upward movement *via* the transpiration stream, reabsorption into shoot xylem parenchyma cells, lateral transfer, loading into and unloading from the phloem (companion cells) for source-to-sink distribution (Figure 1). AtHMA5, a member of the PIB-type ATPase family, is the only transporter identified thus far with the assigned role for Cu loading into the root xylem (Andrés-Colás et al., 2006). In the xylem, Cu is associated with the methionine-derived non-proteinogenic amino acid, nicotianamine (NA), which directly contributes to Cu root-to-shoot translocation in tomato and tobacco, and, as was shown recently, in *A. thaliana* (Chao et al., 2021; Pich & Scholz, 1996; Takahashi et al., 2003).

Historically, NA was primarily linked with iron (Fe) transport, as it is a precursor for Fe(III)-chelating phytosiderophores (PSs) that mediate $Fe^{3+}$ uptake in grasses, while in non-grass species, NA supports long-distance Fe transport mainly *via* the phloem from source to sink tissues (Curie et al., 2009; Stephan & Procházka, 1989). Consistent with this role, the NA-deficient *chloronerva* mutant of tomato displays Fe deficiency symptoms, including interveinal chlorosis, altered root morphology and stunted growth, despite adequate Fe supply (Curie et al., 2009; Stephan & Procházka, 1989). In addition, the *Arabidopsis nax4-2* mutant, lacking all four NA synthase (NAS) genes, is sterile and shows a *chloronerva*-like phenotype (Klatte et al., 2009; Stephan & Grun, 1989). Moreover, the *nax4-1* mutant, which retains residual NA accumulation in leaves, exhibits a marked reduction in seed Fe content, further supporting the role for NA in the phloem-based, long-distance Fe transport (Klatte et al., 2009).

Notably, however, NA forms highly stable complexes with Cu. Based on stability constants (log K) for NA-metal complexes, $Cu^{2+}$ forms one of the most stable complexes with NA, second only to $Fe^{3+}$. The strength of binding decreases in the order: $Fe^{3+}$ (20.6) > $Cu^{2+}$ (18.6) > $Ni^{2+}$ (16.1) > $Co^{2+}$ (14.8) ≈ $Zn^{2+}$ (14.7) > $Fe^{2+}$ (12.1) > Mn (Beneš et al., 1983; Blindauer & Schmid, 2010). This high affinity suggests that NA can readily chelate Cu, supporting the role of NA in long-distance Cu transport as well. Indeed, in tomato, Fe(III) is predominantly chelated by citrate in the xylem, leaving NA available to mobilize Cu(II) (Pich & Scholz, 1996; Rellán-Álvarez et al., 2009). Further support for the role of NA in long-distance Cu transport comes from the discovery of two NA efflux transporters, NAET1 and NAET2, members of the NRT1/PTR

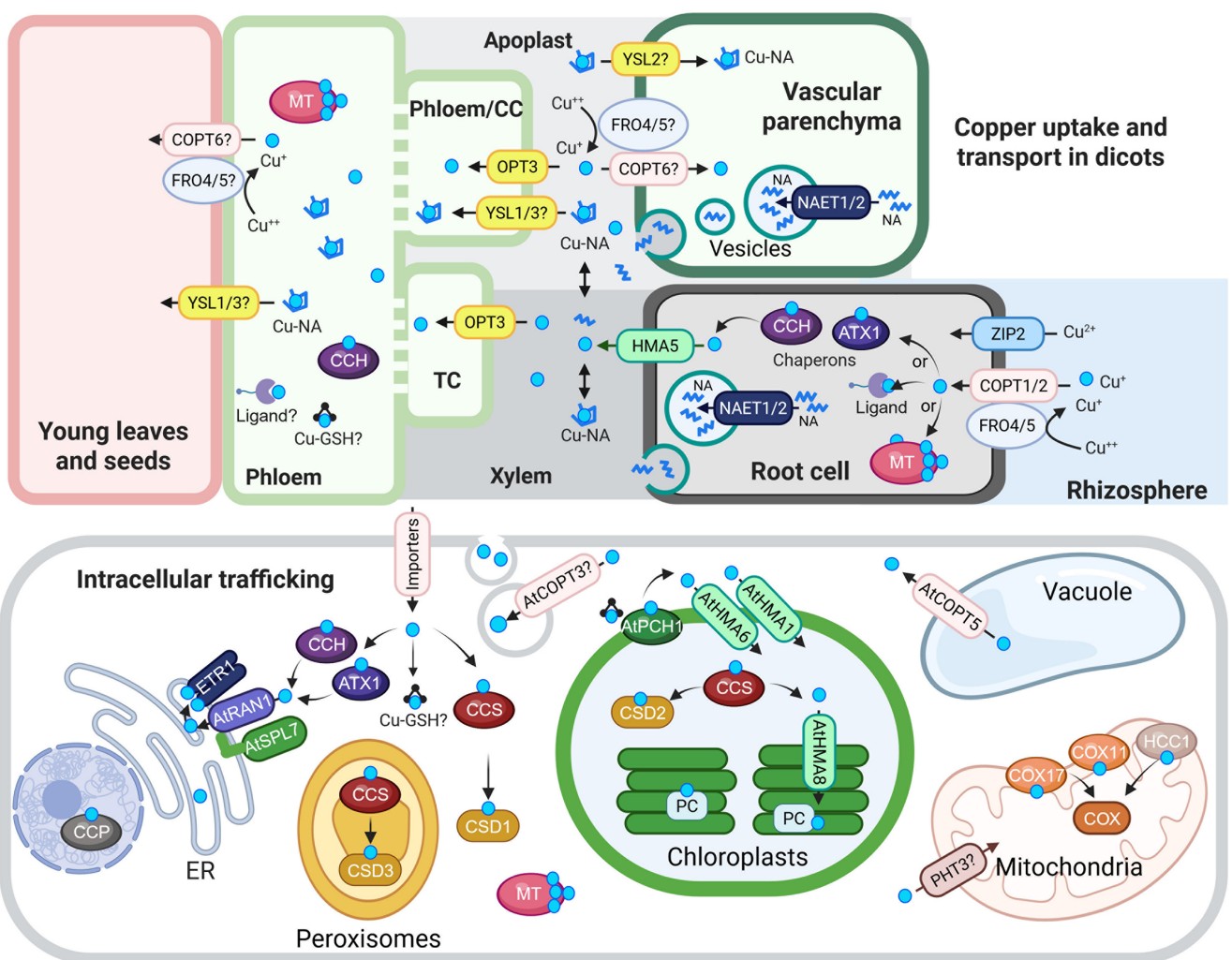

**Figure 1.** Cu uptake, transport and intracellular trafficking in dicots.

**Top panel: Cu uptake and long-distance transport.** At the root surface, Cu(II) is reduced to Cu(I) by AtFRO4 and AtFRO5, then imported into cells via AtCOPT1 and AtCOPT2, members of the CTR/COPT transporter family. AtZIP2 serves as an alternative Cu(II) uptake route. Once inside root cells, Cu is loaded into the xylem via AtHMA5, a member of the HMA family of P-type ATPases. Nicotianamine (NA), synthesized in the cytoplasm, is transported into secretory vesicles by AtNAET1/2, then secreted into the xylem apoplast, where it chelates Cu to form Cu-NA complexes. These complexes are carried to the shoot with the transpiration stream. Cu or Cu-NA may be reabsorbed from the xylem apoplast into xylem parenchyma cells *via* AtCOPT6 and possibly AtYSL2, both expressed in the vasculature and proposed to mediate lateral mineral movement. In the phloem apoplast, Cu or Cu-NA is taken up by AtYSL1/3 or directly by AtOPT3, which mediates Cu loading into phloem companion cells. Although AtFRO4/5 are expressed in the shoot, their role in Cu(II) reduction in the shoot remains unconfirmed. Once in the phloem, Cu transport to sink tissues may involve metallothioneins (MTs), glutathione (GSH) and the Cu chaperone CCH.

**Bottom panel: Intracellular Cu trafficking.** After entering the cytosol, Cu is buffered and distributed by chaperones, ligands, MTs and GSH. At the chloroplast membrane, AtPCH1 delivers Cu to AtHMA1 and AtHMA6, which import Cu into the stroma. There, AtCCS loads Cu onto Cu/Zn-superoxide dismutase (AtCSD2). AtHMA8 then transfers Cu into the thylakoid lumen, where plastocyanin (PC) acquires it. AtPHT3 family transporters may deliver Cu to the mitochondria, and AtCOX11, AtCOX17 and AtHCC1 facilitate its insertion into COX. AtCCS also delivers Cu to CSD1 and CSD3, Cu/Zn-superoxide dismutase isoforms in the cytosol and peroxisomes, respectively. In the vacuole, excess Cu is stored and exported to the cytosol by AtCOPT5, a tonoplast-localized transporter. ATX1 and CCH deliver Cu to RAN1/AtHMA7 on the ER membrane. RAN1/AtHMA7 interacts with AtSPL7; this interaction is speculated to modulate Cu transfer to ETR via RAN1. AtCCP is proposed to be a nuclear Cu chaperone required for plant immunity. Finally, AtCOPT3 associates with secretory vesicles and may participate in Cu mobilization during reproduction.

In the schematic, Cu(I/II) ions are indicated by cyan circles unless otherwise noted. Transporter families are color-coded: CTR/COPT in pink, HMA in green and YSL in yellow. Abbreviations: CC, companion cells; TC, phloem companion cells that de-differentiated into transfer cells; NA, nicotianamine, ER, endoplasmic reticulum; CCS, Cu chaperone for superoxide dismutase; ATX1, antioxidant protein 1; CCH, ATX1-like Cu chaperone; MT, metallothionine; GSH, glutathione; PCH, plastid chaperone 1; PC, plastocyanin. COX, cytochrome c oxidase. ETR1, ethylene response 1. CCP, copper chaperone protein. The figure was created with BioRender.com.

family (Chao et al., 2021). NAET 1 and NAET2 mediate NA secretion into secretory vesicles, which are then transported to the cell surface and released into the apoplast space, creating an NA pool outside the cell for metal binding. Consequently, NA levels are low in the xylem and phloem sap of the *naet1 naet2* double mutant (Chao et al., 2021). These mutants also exhibit phenotypes similar to *chloronerva* in tomato and *nas4x-2* or *ysl1 ysl3* mutants in *A. thaliana*, all of which are defective in NA biosynthesis or

metal-NA transport (Chao et al., 2021; Chu et al., 2010; Klatte et al., 2009; Waters et al., 2006). Interestingly, Fe levels in xylem sap of the *naet1 naet2* double mutant remain unchanged, but Cu concentrations are reduced by nearly 50%. This suggests that, similar to tomato, NA facilitates xylem-based root-to-shoot Cu transport. Concerning phloem sap, the concentration of both Fe and Cu was reduced in the *naet1 naet2* double mutant. Collectively, these findings demonstrate that, in addition to its established role

in Fe transport *via* the phloem, NA also facilitates xylem and phloem-based long-distance transport of Cu.

Concerning Cu transport in the shoot, from what is known thus far, a COPT family member, AtCOPT6, localizes to the plasma membrane and is proposed to mediate lateral Cu transport in above-ground tissues (Jung et al., 2012). Cu reabsorption from the xylem and phloem-based sources-to-sink distribution is mainly mediated by members of the oligopeptide transporter (OPT) family. This family comprises two major clades: the OPTs and yellow stripe-like (YSL) transporters (Lubkowitz, 2011). Of YSLs, AtYSL2 has been shown to transport a Cu-NA in the heterologous yeast system. It is expressed in the vasculature, and its mRNA level increases under Cu deficiency, declining under high Cu treatments (DiDonato et al., 2004). In addition to Cu, AtYSL2 is postulated to transport Fe, but its expression declines under Fe deficiency, suggesting that AtYSL2 contributes to Cu-Fe crosstalk (DiDonato et al., 2004). Based on the expression pattern in the vasculature, it is proposed that AtYSL2 mediates lateral Cu (and Fe) movement in the vascular system, perhaps *via* reabsorption from the xylem into the parenchyma cells. AtYSL1 and AtYSL3, on the other hand, localize to phloem companion cells and contribute to Cu remobilization to reproductive tissues and seeds (Chu et al., 2010; Mustroph et al., 2009; Waters et al., 2006). In addition to resembling the tomato *chloronerva* mutant, *ysl1 ysl3* double mutants show reduced Cu levels in seeds and flowers, delayed flowering and reduced fertility, phenotypes that are partially rescued by Cu supplementation (Chu et al., 2010; Waters et al., 2006).

Cu loading into the phloem companion cells is mediated by AtOPT3, a member of the OPT clade of the OPT family (Chia et al., 2023). The *opt3-3* mutant exhibits reduced Cu in phloem sap and impaired source-to-sink Cu movement, indicating the role of AtOPT3 in Cu phloem loading and redistribution to young tissues and developing seeds (Chia et al., 2023; Zhai et al., 2014). AtOPT3 was originally identified as a critical component of Fe homeostasis and root-to-shoot signalling in *A. thaliana* (Stacey et al., 2007) and detailed in Section 4.2. The transport substrate of AtOPT3 has been debated, with hypotheses ranging from peptide-metal complexes to free metal ions. Indeed, early studies show that members of the OPT and YSL clades of the OPT family transport tetra- and pentapeptides and metal-NA complexes, respectively (Curie et al., 2009; Osawa et al., 2006). More recent evidence has shown that AtOPT3 can transport free $Fe^{2+}$, $Cd^{2+}$ and $Cu^{2+}$ ions when expressed in *Xenopus* oocytes and yeast metal transport mutants (Chia et al., 2023; Zhai et al., 2014). Notably, $Cu^{2+}$ was the preferred substrate over Cu-NA in oocyte assays (Chia et al., 2023). In parallel, NAET1 and NAET2, recently identified as NA exporters, release NA into the phloem companion cells, influencing Fe and Cu delivery to seeds (Chao et al., 2021). These findings suggest that AtOPT3 and NAET1/2 may act in concert to facilitate the co-delivery of free $Cu^{2+}$ and $Fe^{2+}$, and their complexation with NA in the phloem, enabling efficient long-distance transport.

In addition to NA, other molecules contribute to Cu transport and redistribution in plants. Metallothioneins (MTs), cysteine-rich proteins, bind Cu through thiol groups and function in buffering excess Cu as well as in its redistribution from leaves to seeds (Benatti et al., 2014; Guo et al., 2008). Glutathione (GSH), a ubiquitous tripeptide, also chelates Cu and facilitates Cu uptake and trafficking in animals, while its role in plants is mainly associated with detoxification of excessive Cu concentrations (Burkhead et al., 2009; Maryon et al., 2013). Whether GSH is also involved in Cu trafficking and/or long-distance transport in plants needs to be verified experimentally. The ATX1-like Cu chaperone, CCH, is detected in the phloem and has been implicated in Cu loading and long-distance distribution, ensuring adequate supply to sink tissues (Himelblau et al., 1998; Mira et al., 2001). Thus, Cu transport involves not only transporters but also a network of Cu-binding peptides and chaperones.

### 2.2. Cu uptake and long-distance transport in grasses

Copper uptake strategies in grasses remain less well understood than in dicots. Moreover, studies on the ionic form of Cu absorbed by grass roots have yielded conflicting results, suggesting three not mutually exclusive models for Cu uptake (Figure 2):

1) *Uptake of $Cu^+$ (Cu(I))*: Evidence from (Jouvin et al., 2012) suggests that, as in non-graminaceous plants, Cu(II) is reduced to Cu(I) before uptake into the root epidermal cells of grasses. This implies the existence of root surface reductase activity in grasses, analogous to FRO3/4 enzymes in *A. thaliana* and other dicots (Figures 1 and 2). However, unlike non-grass species, which possess multiple FRO-like genes, grasses such as corn, *Brachypodium distachyon* (hereafter *brachypodium*), and rice harbour only one or two *FRO*-like genes (Table 1; (Schwacke et al., 2003; Schwacke & Flügge, 2018). Still, the limited number of *FRO* genes does not exclude the possibility that grasses use an as-yet-unidentified mechanism for Cu(II) reduction at the root surface. As for Cu(I) transporters, several COPT/CTR family members in rice, including OsCOPT1, OsCOPT5, OsCOPT6 and OsCOPT7, are transcriptionally upregulated in roots under Cu deficiency (Yuan et al., 2011). OsCOPT1 and OsCOPT5 have been shown to contribute to Cu accumulation in the shoots, but whether these transporters mediate Cu uptake is unknown (Yao et al., 2022). Interestingly, however, loss of their function not only decreased shoot Cu levels but also reduced viral resistance, exemplifying the role of Cu in pathogen resistance (Yao et al., 2022). In *brachypodium*, five COPT genes have been identified, with BdCOPT3 and BdCOPT4 being Cu-deficiency-inducible in both roots and shoots. These proteins localize to the plasma membrane but, unlike their high-affinity *A. thaliana* counterparts, they mediate low-affinity Cu uptake, suggesting mechanistic divergence in Cu acquisition from *A. thaliana* (Jung et al., 2014).

2) *Uptake of $Cu^{2+}$ (Cu(II))*: Alternatively, Ryan et al. (2013) found that roots of oat (*Avena sativa*) absorb Cu(II) directly from the soil solution, with Cu(II) subsequently reduced to Cu(I) within root cells. This model implies the existence of Cu(II) transporters at the plasma membrane (Figure 2). While ZIP2 has been recognized for its function in mediating Cu uptake, presumably as Cu(II), in the roots of *A. thaliana* (Robe et al., 2025; Wintz et al., 2003), corresponding ZIP transporters in grasses remain unidentified. Recently, ZmYS1 was found to mediate Cu(II) transport when expressed in *Xenopus* oocytes. Although ZmYS1 is a well-characterized metal-NA and metal-PS transporter, ZmYS1-expressing oocytes accumulated significantly higher levels of Cu when it was supplied in ionic form compared to the Cu-NA complex (Sheng et al., 2021). The physiological substrate specificity of ZmYS1 and the contribution of YSL family members to Cu(II) uptake *in planta* remain to be established.

3) *Uptake of Cu(II)-ligand complexes*: A third scenario involves the uptake of Cu complexed with ligands, particularly NA, that strongly chelates Cu(II) and plays a known role in long-distance Cu transport in both grasses and dicots (reviewed in (Curie et al., 2009)). Whether NA is secreted into the rhizosphere by Cu-deficient roots to mobilize soil Cu is unknown. Grasses are known

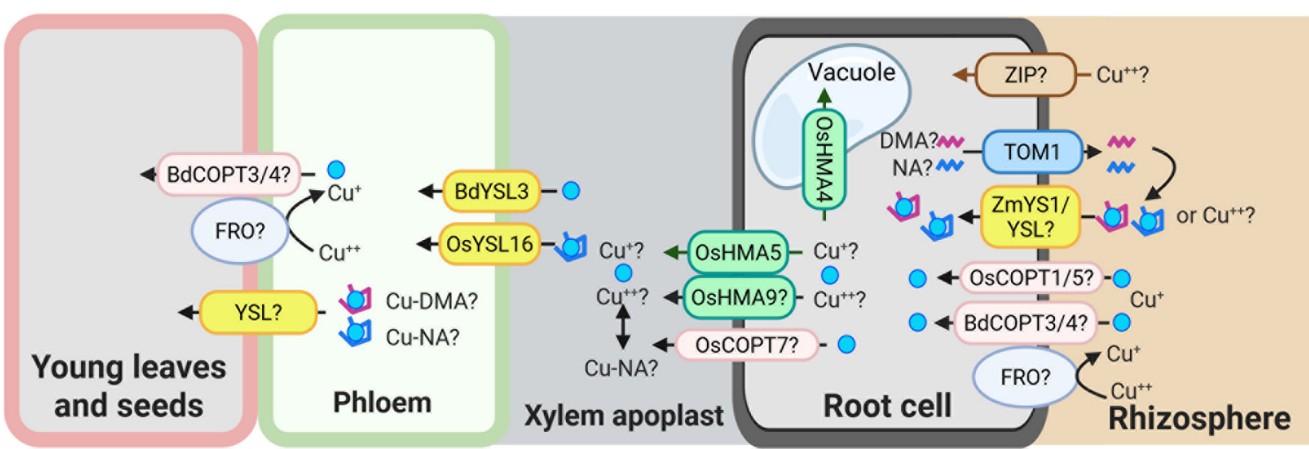

**Figure 2.** Speculative model for Cu uptake and transport in grass species.

This model illustrates potential mechanisms of Cu uptake and internal transport in grasses. Cu(II) uptake from the rhizosphere may involve as-yet unidentified ZIP-like transporters and/or YS1/YSL family members. Cu uptake may also occur via secretion of phytosiderophores, primarily 2*r*-deoxymugineic acid (DMA), and possibly nicotianamine (NA), through the TOM1-like transporters. The resulting Cu-DMA or Cu-NA complexes can then be imported into root cells *via* ZmYS1/YSL transporters. A low-affinity Cu uptake system, potentially involving BdCOPT3/4, OsCOPT1/5 and an FRO-type reductase, may also contribute to Cu acquisition directly from the rhizosphere.

Once inside the root, Cu is loaded into the xylem by HMA family transporters, including OsHMA5 and possibly OsHMA9 and OsCOPT7. OsHMA4 was identified as a vacuolar Cu importer in rice. The transfer of Cu from xylem to phloem is mediated by BdYSL3 and OsYSL16, both of which facilitate Cu redistribution towards sink tissues. Although the precise mechanisms of Cu unloading into sink organs remain to be elucidated, available evidence suggests that YSL transporters and components of the low-affinity Cu uptake pathway may play a role in this process as well. The figure was created with BioRender.com.

to secrete PSs to solubilize Fe(III) under Fe deficiency, and this process involves TOM1 (Transporter of Mugineic Acid PSs) in rice and barley (Nozoye et al., 2011; Takagi, 1976). As it was shown in maize, YS1 mediates Fe-PC complexes uptake into root cells (Curie et al., 2001). Notably, ZmYS1 is capable of transporting Cu-PS and Cu-NA complexes in the heterologous system (Chia et al., 2023; Curie et al., 2001; Schaaf et al., 2004). However, whether NA, MAs and YS1/YSLs are involved in Cu uptake *in planta* remains an open question.

Concerning Cu long-distance transport in grasses, several homologs of AtHMA5 in rice, including OsHMA5 and possibly OsHMA9, are induced by high Cu and have been proposed to mediate Cu loading into the root xylem, thereby contributing to long-distance transport (Andrés-Colás et al., 2006; Deng et al., 2013; Kobayashi et al., 2008; Lee et al., 2007). OsHMA6 also belongs to the AtHMA5 clade and shares 83% sequence similarity with OsHMA9 (Zou et al., 2020). It has been demonstrated to function as a plasma membrane Cu efflux transporter, although it remains unclear whether OsHMA6 also participates in the long-distance transport of Cu in rice. More recently, OsCOPT7 was demonstrated to be predominantly expressed in the exodermis and stele of roots, facilitating root-to-shoot translocation *via* xylem, and contributing to Cu accumulation in rice grains (Guan et al., 2024).

Among the YSL transporters, OsYSL16, transports Cu-NA complexes when expressed heterologously in yeast lacking high-affinity COPTs CTR1p and CTR3p (Zheng et al., 2012). Loss-of-function mutations in OsYSL16 have a defect in Cu redistribution from sources, such as mature leaves and developing flag leaves, to sink tissues, including panicles; this defect results in reduced grain yield (Zheng et al., 2012). Its ortholog in *brachypodium*, BdYSL3, is transcriptionally upregulated under Cu deficiency and mediates Cu redistribution from mature leaves to flag leaves, reproductive organs, and developing grains as well (Sheng et al., 2021). Notably, BdYSL3 mutants exhibit decreased seed Cu levels, reduced seed size and diminished protein content, directly linking Cu transport to

the development of these key agronomic traits. It has been shown that YSL transporters from different species mediate the transport of Cu-NA or Cu-PSs complexes (DiDonato et al., 2004; Schaaf et al., 2004; Zheng et al., 2012). However, analysis of the transport capabilities of BdYSL3 using the *Xenopus laevis* oocyte expression system has shown that oocytes expressing *BdYSL3* accumulate Cu only when it is supplied in ionic form, but not as the Cu-NA complex (Sheng et al., 2021). By contrast, oocytes expressing ZmYS1 accumulate Cu when it is supplied as Cu-NA complex; however, Cu levels in ZmYS1-expressing oocytes increase significantly when Cu is supplied in the ionic form (Sheng et al., 2021). The physiological substrate(s) of YSLs are yet to be determined.

### 2.3. Intracellular Cu transport and trafficking

The cytosol maintains extremely low concentrations of free Cu ions, estimated to be in the femtomolar to zeptomolar range, through tight buffering by cellular ligands and Cu chaperones (Ackerman et al., 2017; Hong-Hermesdorf et al., 2014; Schmollinger et al., 2021; Waldron & Robinson, 2009; Wintz & Vulpe, 2002). These ligands and chaperones ensure safe and efficient delivery of Cu to its target proteins and organelles. The *A. thaliana* genome encodes nine Cu chaperones: Cu chaperone for superoxide dismutase (CCS), antioxidant protein 1 (ATX1), ATX1-like chaperone (CCH), cytochrome c oxidase 11 (COX11), COX17, two homologs of the yeast Cu chaperone (HCC1 and HCC2) and plastid Cu chaperone 1 (PCH1) (Attallah et al., 2011; Balandin & Castresana, 2002; Blaby-Haas et al., 2014; Chu et al., 2005; Himelblau et al., 1998; Llases et al., 2020; Shin et al., 2012; Steinebrunner et al., 2011). A new putative Cu chaperone induced by pathogens (CCP) has been recently identified in *A. thaliana* that, unlike other chaperones, localizes to the nucleus (Chai et al., 2020). Most of these proteins share a conserved heavy metal-binding domain, characterized by an MXXCXC motif, and function through protein-protein interactions, guiding Cu from cellular entry points to destination cuproenzymes or

**Table 1.** A list of *FRO*-like genes in monocot and dicot model plants. The accession numbers of the FRO-homologs were retrieved from the Arabidopsis Plant Membrane Protein database (Aramemnon) (Schwacke et al., 2003; Schwacke & Flügge, 2018).

| Plant species | Accession numbers | Name |
|---|---|---|
| *Arabidopsis thaliana* | At1g01590 | Putative ferric-chelate reductase (*AtFRO1*) |
| | At1g01580 | Putative ferric-chelate reductase (*AtFRO2*) |
| | At1g23020 | Putative ferric-chelate reductase (*AtFRO3*) |
| | At5g23980 | Putative ferric-chelate reductase (*AtFRO4*) |
| | At5g23990 | Putative ferric-chelate reductase (*AtFRO5*) |
| | At5g49730 | Putative ferric-chelate reductase (*AtFRO6*) |
| | At5g49740 | Putative ferric-chelate reductase (*AtFRO7*) |
| | At5g50160 | Putative ferric-chelate reductase (*AtFRO8*) |
| *Populus trichocarpa* | Potri.017G142700 | Putative ferric-chelate reductase |
| | Potri.T061300 | Putative ferric-chelate reductase |
| | Potri.004G079100 | Putative ferric-chelate reductase |
| | Potri.017G142800 | Putative ferric-chelate reductase |
| | Potri.012G084800 | Putative ferric-chelate reductase |
| | Potri.001G079000 | Putative ferric-chelate reductase |
| | Potri.004G079200 | Putative ferric-chelate reductase |
| | Potri.014G088000 | Putative ferric-chelate reductase |
| | Potri.015G083200 | Putative ferric-chelate reductase |
| *Solanum lycopersicum* | Solyc01g094910 | Putative ferric-chelate reductase (*LeFRO1*) |
| | Solyc01g094890 | Putative ferric-chelate reductase |
| | Solyc06g036220 | Putative ferric-chelate reductase |
| | Solyc00g026160 | Putative ferric-chelate reductase |
| | Solyc01g094900 | Putative ferric-chelate reductase |
| | Solyc03g112320 | Putative ferric-chelate reductase |
| | Solyc01g102610 | Putative ferric-chelate reductase |
| *Brachypodium distachyon* | Bradi5g11147 | Putative ferric-chelate reductase |
| | Bradi5g19150 | Putative ferric-chelate reductase |
| *Oryza sativa* | Os04g36720 | Putative ferric-chelate reductase (*OsFRO1*) |
| | Os04g48930 | Putative ferric-chelate reductase (*OsFRO2*) |
| *Zea mays* | GRMZM2G068557 | Putative ferric-chelate reductase |

transporters within the intracellular compartments (Figure 1) and (Harrison et al., 1999; Rono et al., 2022). For example, ATX1 mediates Cu transfer to rice HMA4, HMA5, HMA6, HMA9 and COPT7, facilitating root-to-shoot translocation and redistribution of Cu from old leaves to developing tissues and seeds (Guan et al., 2024; Zhang et al., 2018). Similarly, CCS, delivers Cu to three types of Cu/Zn Superoxide Dismutases: CSD1, in the cytosol, CSD2 in the chloroplast and CSD3 in peroxisomes (Bowler et al., 1992; Chu et al., 2005; Kliebenstein et al., 1998). CSD1 is a major Cu sink in the cytosol, and mRNA abundance for both *CCS* and *CSD1* (and as described below, *CDS2*) negatively correlates with Cu availability in a microRNA 398 (miR398)-dependent manner (Cohu et al., 2009).

*Chloroplasts* are the main sinks for copper (Cu) in photosynthetic tissues, making up an estimated 30% of the overall cellular Cu in mature leaves, as was estimated for soil-grown *A. thaliana* (Shikanai et al., 2003). This is primarily because Cu is needed for plastocyanin, an essential soluble electron carrier that functions between the cytochrome-*b6f* complex and photosystem I (Burkhead et al., 2009; Weigel et al., 2003; Yruela, 2009). Plant genomes contain two plastocyanin isoforms, PC1 and PC2, which have identical roles in electron transport (Abdel-Ghany, 2009). Another major Cu-containing protein in the chloroplast is CSD2, which requires a Cu chaperone, CCS, for Cu delivery (Cohu et al., 2009). A dedicated chaperone inserting Cu into plastocyanin has not yet been identified.

Additionally, many plants, including poplar and spinach, have the thylakoid lumen associated class of Cu enzymes, polyphenol oxidases (PPOs), which oxidize a variety of aromatic compounds. These enzymes are believed to play roles in plant defence and in synthesizing specialized metabolites. However, *A. thaliana* and green algae like *Chlamydomonas reinhardtii* do not have PPOs in this lumen (Mayer, 2006; Sullivan, 2014).

Cu likely enters the chloroplast intermembrane space by diffusion through porins in the outer membrane, potentially bound to GSH or a PCH1 (Aguirre & Pilon, 2015; Blaby-Haas et al., 2014). In *A. thaliana*, PCH1 delivers Cu to a PIB-type Cu-transporting ATPase, PAA1 (Plastid ATPase of Arabidopsis 1, *alias* HMA6), localized to the inner envelope and actively

transporting Cu into the chloroplast stroma (Shikanai et al., 2003). *PAA1* loss-of-function mutants are chlorotic due to a defect in plastocyanin assembly; this phenotype underlies the importance of PAA1/HMA6 transporter in chloroplast function. Another PIB-ATPase family member, AtHMA1, localizes to the chloroplast and is involved in chloroplast Cu homeostasis, as evidenced by the decreased chloroplast Cu levels in the *hma1* mutants (Boutigny et al., 2014). Unlike *hma6* mutans, loss of AtHMA1 did not cause obvious phenotypes under standard conditions; however, the *hma1* mutants are susceptible to high light.

Once in the stroma, Cu is further delivered to the thylakoid lumen by another PIB-type ATPase, PAA2 (*alias* HMA8), localized to the thylakoid membrane (Abdel-Ghany et al., 2005). PAA2/HMA8 moves Cu from the stroma into the thylakoid lumen, where it is incorporated into plastocyanin (Figure 1). Together, the sequential actions of Cu transporters and chaperones ensure the proper metalation of key Cu-dependent proteins in the chloroplast while preventing Cu toxicity (Abdel-Ghany et al., 2005; Seigneurin-Berny et al., 2006; Shikanai et al., 2003).

*Mitochondria*, a second major Cu sink, contain 5-15% of total cellular Cu, used primarily by cytochrome *c* oxidase (COX, or mitochondrial complex IV), which transfers electrons from cytochrome *c* to oxygen as the terminal electron acceptor of the respiratory mitochondrial electron transport chain (Kadenbach et al., 2000; Ravet & Pilon, 2013). *A. thaliana* mitochondrial Cu pools are maintained by chaperones, such as COX11, COX17 and HCC1, all essential for COX assembly and mitochondrial respiration (Attallah et al., 2011; Burkhead et al., 2009; Llases et al., 2020; Steinebrunner et al., 2014). HCC1 homolog, HCC2, localizes to the mitochondria as well, but lacks canonical Cu binding motifs and is implicated in response to UV-B radiation (Steinebrunner et al., 2014).

Cu transport in and out of mitochondria in plants is poorly understood despite the essential role of this metal in mitochondrial function. Concerning other species, the mitochondrial carrier family (MCF) member in *Saccharomyces cerevisiae*, Pic2, and its homolog in humans, SLC25A3, import Cu into the mitochondrial matrix (Boulet et al., 2018; Vest et al., 2013; Zhu et al., 2021). MCF-like proteins are also found in plants. Three orthologous genes have been identified in *A. thaliana* genome and designated as AtMPT1/AT2/PHT3.3, AtMPT2/AT3/PHT3.2 and AtMPT3/AT5/PHT3.1 (Hamel et al., 2004; Poirier & Bucher, 2002). Previous studies suggested that they participate in phosphate transport (Poirier & Bucher, 2002; Zhu et al., 2012). Whether any of them is involved in Cu import into mitochondria is unknown.

The *vacuole* serves as a dynamic storage compartment for Cu, particularly under conditions of excess. Plant transporters mediating vacuolar Cu sequestration are not clearly defined. AtCOPT3 localizes to the secretory pathway and is expressed in the vasculature and pollen grains, and its overexpression increases plant Cu levels (Andrés-Colás et al., 2018). AtCOPT5 localizes to the tonoplast and mediates Cu efflux under Cu deficiency (Andrés-Colás et al., 2010). This process allows plants to buffer cytosolic Cu levels and remobilize stored Cu when needed, as evidenced by the compromised growth of the *copt5* mutants under Cu deficiency (Garcia-Molina et al., 2011; Klaumann et al., 2011). Consistent with the notion of the existing crosstalk between Cu and Fe homeostasis, *copt5* mutants are sensitive to Fe deficiency, while the loss of Fe exporters NRAMP3 and NRAMP4 increases the plant's sensitivity to Cu deficiency (Carrió-Seguí et al., 2019).

In rice, OsHMA4 and its transcript variants function in sequestering Cu into root vacuoles, thereby limiting Cu accumulation in the grain (Guan et al., 2022; Guan et al., 2024). Heterologous expression studies further demonstrated that OsHMA4 localizes to the tonoplast and enhances vacuolar Cu sequestration. Loss-of-function mutants of OsHMA4 exhibit elevated Cu in the shoots, xylem sap and grains, confirming its role as a vacuolar Cu importer in cellular Cu homeostasis and maintaining root-to-shoot Cu balance.

The *nucleus* and *endoplasmic reticulum (ER)* also harbour small but functionally critical pools of Cu. RESPONSIVE-TO-ANTAGONIST 1 (RAN1, alias AtHMA7) transports Cu across the ER membrane to provide Cu(I) to the transmembrane sensor domain of the ethylene receptor ETHYLENE RESPONSE 1 (ETR1) to confer proper ethylene response (Binder et al., 2010; Hirayama et al., 1999; Rodríguez et al., 1999; Woeste & Kieber, 2000) (Figure 1). Cu chaperon ATX1 traffics Cu to RAN1 to be incorporated to ETR1 and the partial loss-of-function *Arabidopsis ran1-1* mutant exhibits a mild constitutive ethylene response (Li et al., 2017; Rodríguez et al., 1999). High-affinity ethylene binding by ETR1 is primarily mediated by Cu(I) coordinated to a conserved cysteine and histidine residues in the second transmembrane helix of ETR N-terminal ethylene-binding domain, while an asparagine residue plays a crucial role in positioning the histidine residue for Cu coordination, further contributing to the high-affinity binding (Azhar et al., 2023; Rodríguez et al., 1999; Schaller et al., 1995). Structural modelling of the transmembrane sensor domain of *Arabidopsis* ETR revealed a dimeric state with two Cu(I) per receptor dimer and potentially two ethylene-binding sites per receptor dimer (Schott-Verdugo et al., 2019).

*Peroxisomes*, multifunctional organelles that contain enzymes catalysing diverse oxidative reactions, require Cu to manage oxidative stress (Corpas et al., 2020). For example, CSD3 (Cu/Zn-SOD) is a major Cu protein in peroxisomes, and, as noted above, Cu is delivered to CSD3 *via* a Cu chaperone CCS (Chu et al., 2005). In addition, peroxisomes possess Cu-containing amine oxidases, enzymes that oxidize polyamines with concomitant production of $H_2O_2$ and $NH_3$ (Planas-Portell et al., 2013). How Cu is delivered to peroxisomes is unknown.

The role of Cu and its precise localization in the *nucleus* of plants is not fully understood as well. However, a recently discovered Cu chaperone induced by pathogens (CCP), contains a nuclear localization signal (Chai et al., 2020). CCP interacts with transcription factor TGA2 *in vivo* and *in vitro* and induces SA-mediated defence signalling (Chai et al., 2020). Recent reports from the studies in animal systems indicate that Cu ions are predominantly located in the perinuclear and nuclear areas, including nucleoli, and act during mitosis; specifically, Cu binding to cyclins and subsequent antioxidant 1 chaperon-dependent transfer to cyclin-dependent kinase 1 is essential for its kinase function, triggering the G2/M progression in the cell cycle (McRae et al., 2013; Wang et al., 2025; Yang et al., 2005). A newly discovered function of the recombinant *Xenopus laevis* histone H3-H4 tetramer in reducing Cu(II) to Cu(I), and the essentiality of this function of the H3 histones for Cu utilization in *S. cerevisiae*, is a testament to the untapped Cu cellular resources and roles in biology (Attar et al., 2020).

## 3. Regulation of Cu homeostasis

Plants employ two major strategies to maintain Cu homeostasis: regulation of Cu acquisition and the Cu economy/metal switch mechanism (Figure 3A). In *A. thaliana*, Cu acquisition is regulated by the conserved transcription factor SQUAMOSA

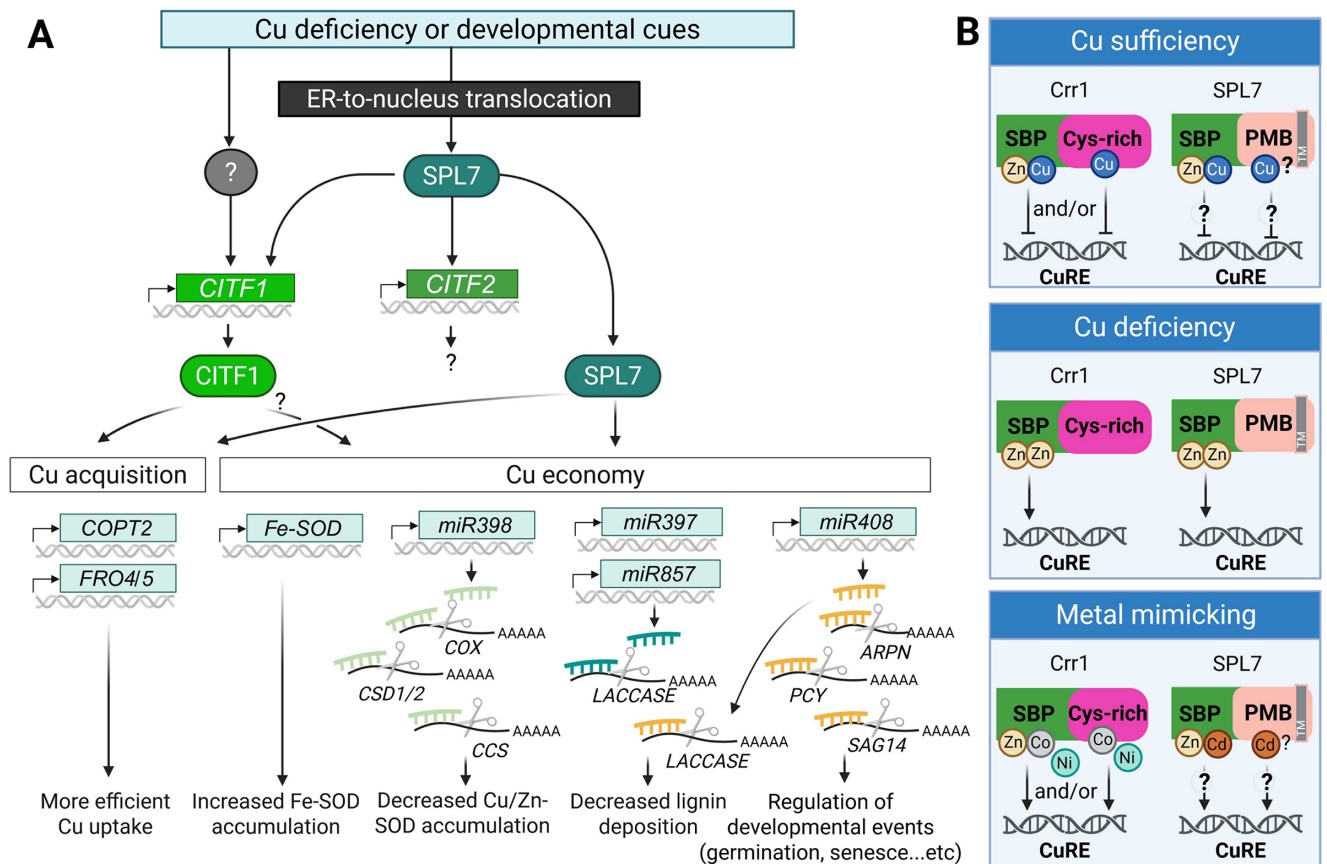

**Figure 3.** Summary of transcriptional Cu deficiency response based on studies in *A. thaliana*.
(A) Cu deficiency responses are triggered by fluctuations in environmental Cu availability and/or developmental cues. The master regulator SPL7 is translocated from the endoplasmic reticulum (ER) membrane to the nucleus, where it activates a Cu deficiency response cascade. This includes the induction of *CITF1*, *CITF2*, and downstream genes involved in Cu acquisition and Cu economy. *CITF1* expression is partially dependent on SPL7 and may also be regulated by yet unidentified transcription factors. SPL7 and CITF1 co-regulate several Cu uptake-related genes, including *COPT2*, *FRO4* and *FRO5*. The Cu economy is primarily governed by SPL7, and potentially by CITF1. Under Cu-deficient conditions, *FeSOD* (*FSD1*) expression is upregulated to substitute for *Cu/Zn-SOD*, *CSD1/2*. A set of Cu-responsive microRNAs is also activated to degrade mRNAs encoding Cu-dependent proteins, such as *CSD1/2*, *COX*, *CSS*, *Laccases*, *ARPN* and a pair of interacting plantacyanin proteins (plantacyanin [*PCY*] and senescence-associated gene 14 [*SAG14*]), thereby reducing overall Cu consumption.
(B) Speculative models for AtSPL7/Crr1 Cu status sensing (based on Gayomba et al., 2013; Kropat et al., 2005). Under Cu sufficiency, Cu may replace Zn in the SBP domain in Crr1 and AtSPL7 and/or bind the C-terminal Cys-rich region in Crr1, altering the transcription factor's structure to prevent binding to the Cu response element (CuRE). Under Cu deficiency, Zn may replace Cu in the SBP domain and/or induce structural changes that promote CuRE binding. Excess of other divalent metal ions (e.g., $Cd^{2+}$, $Ni^{2+}$ and $Co^{2+}$) may mimic Zn function and also facilitate interaction with CuRE. While the Crr1's Cys-rich domain is not conserved in AtSPL7, the C-terminal half of AtSPL7 contains four potential metal-binding (PMB) sites in addition to multiple Cys and His residues. Their role in metal binding and Cu sensing is yet to be determined. A grey box labelled (TM) in SPL7 indicates the transmembrane domain.
The figure was created with BioRender.com.

PROMOTER BINDING PROTEIN-LIKE 7 (SPL7) and its downstream target, the clade Ib basic helix–loop–helix (bHLH) transcription factor COPPER DEFICIENCY-INDUCED TRANSCRIPTION FACTOR 1 (CITF1, also known as bHLH160) (Bernal et al., 2012; Yamasaki et al., 2009; Yan et al., 2017). Together, SPL7 and CITF1 coregulate genes encoding the high-affinity Cu uptake system in roots, including *COPT2*, *FRO4* and *FRO5*. Although CITF1's direct targets have yet to be identified, chromatin immunoprecipitation followed by deep sequencing (ChIP-seq) has identified *CITF2* (*alias* bHLH23), with an unknown role in Cu homeostasis, and Cu transporters *ZIP2* and *YSL2* as direct SPL7 targets (Schulten et al., 2022).

SPL7 also governs the Cu economy/metal switch mechanism, which prioritizes Cu use for essential functions under Cu deficiency. This mechanism involves miRNAs that target genes encoding abundant and functionally redundant Cu proteins. Their depletion is compensated by the isoforms that utilize other metals, most commonly iron (Fe) (Burkhead et al., 2009). For example, *miR398*, a direct SPL7 target, is transcriptionally upregulated under Cu deficiency and targets *CSD1* and *CSD2*, and their chaperone *CCS*. At the same time, the expression of *FSD1*, an Fe-dependent isoform and direct SPL7 target, increases (Cohu & Pilon, 2007; Schulten et al., 2022; Yamasaki et al., 2007; Yamasaki et al., 2009). Other SPL7-regulated miRNAs include *miR397b*, *miR408*, *miR853a*, *miR857a* and *miR860a*. These miRNAs collectively downregulate genes encoding several non-essential or less critical Cu proteins, including CSDs, CCS and COX subunit 5b (miR398); laccases (miR397, miR408 miR857); plantacyanins, ARPN and PCY and senescence-associated proteins such as SAG14 (miR408). This process allows for prioritizing Cu allocation to essential cellular functions, a regulatory mechanism often referred to as the

Cu economy/metal switch model (Bernal et al., 2012; Cohu et al., 2009; Hao et al., 2022; Pilon, 2017; Sunkar et al., 2006; Yamasaki et al., 2007; Yamasaki et al., 2009).

While *CITF1* transcript levels increase under Cu deficiency and decrease under sufficiency, partially in an SPL7-dependent manner, *AtSPL7* itself is expressed constitutively, suggesting its posttranscriptional regulation under Cu deficiency (Shikanai et al., 2003; Yan et al., 2017). Unlike other SPL family members, the *A. thaliana* SPL7 contains a 20-amino-acid C-terminal transmembrane domain (TMD) in addition to the characteristic SBP domain with a functional bipartite nuclear localization signal (Garcia-Molina et al., 2014). Under Cu-sufficient conditions, the TMD anchors AtSPL7 to the ER, preventing its nuclear localization and transcriptional activity. Under Cu deficiency, AtSPL7 undergoes proteolytic cleavage, releasing the SBP domain-containing N-terminal fragment, which translocates to the nucleus to activate target genes. Furthermore, this N-terminal fragment can homodimerize, forming complexes too large to pass through the nuclear pore, thereby offering an additional regulatory layer over AtSPL7 activity (Garcia-Molina et al., 2014).

AtSPL7, anchored *via* TMD to ER, also interacts with RAN1, a Cu transporter that is required for Cu transfer to ethylene receptors; the TMD of AtSPL7 is required for regulating RAN1 abundance and ethylene sensitivity (Yang et al., 2022) and (Figure 1). Whether and how AtSPL7-RAN1 interaction modulates RAN1-based Cu delivery to ethylene receptors is yet to be determined.

The molecular components regulating Cu homeostasis in grasses remain poorly defined, but recent findings indicate that rice exhibits a partially conserved Cu economy strategy while also employing a conserved SPL7-like system to orchestrate the Cu deficiency response (Navarro et al., 2021). For example, the expression of Cu-miRNA homologs in rice, such as *OsmiR397*, *OsmiR398b* and *OsmiR408*, is significantly upregulated under Cu deficiency, whereas the expression of *CSD* homologs (*OsCSD2/3/4*) is downregulated. However, Cu deficiency does not upregulate *OsFSD* genes or suppress the expression of *OsCCS*, which highlights the unique features of the Cu economy response in rice (Navarro et al., 2021).

The regulatory network of Cu homeostasis in grasses is further depicted by functional analyses of OsSPL9, the closest rice homolog of AtSPL7 (Wang et al., 2024). Loss-of-function mutants of *OsSPL9* display increased sensitivity to Cu deficiency, reduced Cu accumulation in shoots, and impaired Cu distribution to newly developing leaves, underscoring its importance for Cu resilience. OsSPL9 directly binds to GTAC motifs in the promoters of multiple Cu uptake and transport genes, including *OsCOPT1/5/6/7* and *OsYSL16*, as well as genes encoding Cu-responsive miRNAs such as *OsmiR397*, *OsmiR398*, *OsmiR408* and *OsmiR528*, thereby promoting Cu uptake, root-to-shoot translocation, and the Cu economy response (Bai et al., 2025; Wang et al., 2024). Together, these findings position OsSPL9 as a functional equivalent of *Arabidopsis* SPL7 and highlight the conservation of SPL7-based Cu regulatory modules across monocots and dicots.

## 4. Cu sensing and signalling mechanisms

### 4.1. Local sensing of Cu availability

Our understanding of Cu sensing originates from studies in bacteria and green algae, *C. reinhardtii*. In *Escherichia coli*, cytoplasmic Cu sensors like CueR, CsoR and CopY modulate DNA binding in response to Cu ions (Ma et al., 2009; Stoyanov et al., 2001).

When the cytosolic Cu level increases, metal binding to the sensor induces changes in the DNA binding region to promote or repress gene expression. These sensors have a high affinity for Cu ions and are highly sensitive to fluctuations of labile Cu concentration in the cells (Ma et al., 2009; Osman et al., 2019). For example, the Cu binding affinity of CueR is $10^{-21}$ M (Changela et al., 2003). *E. coli* also senses periplasmic Cu through two-component systems, for example, CusS–CusR (Outten et al., 2001). CusS is a homodimeric histidine kinase sensor; CusR is the cognate cytosolic response regulator and has an N-terminal receiver domain that interacts with CusS and a C-terminal effector domain that binds DNA (Affandi et al., 2016; Fu et al., 2020). The equivalent systems in plants have not yet been discovered.

Early insights on Cu sensing in plants came from studies in *C. reinhardtii*, where Cu-response elements (CuREs) with GTAC core motifs were found in promoters of Cu-responsive genes (Kropat et al., 2005; Quinn et al., 2000; Sommer et al., 2010). These motifs are recognized by the Cu response regulator Crr1, a functional ortholog of AtSPL7 (Kropat et al., 2005; Sommer et al., 2010). Recent global ChIP-seq analysis refined the AtSPL7 recognition sites and supported the consensus GTACTRC motif in promoter regions of AtSPL7 targets primarily under low-Cu conditions (Schulten et al., 2022).

Both Crr1 and AtSPL7 contain a Zn-binding SBP domain, regarded as a putative metal-sensing region in Crr1 (Kropat et al., 2005; Yamasaki et al., 2009) and (Figure 3B). In addition, Crr1 has four putative metal-binding CXXC motifs in the C-terminal region. These motifs are not conserved in AtSPL7 (Kropat et al., 2005; Yamasaki et al., 2009). Crr1 binds to GTAC motifs via the SBP domain in a Zn-dependent manner under Cu deficiency. It is proposed that under sufficiency, Cu may displace Zn in the SBP domain and/or induce conformational changes *via* the C-terminal Cys-rich region, releasing Crr1 from the promoters of its targets (Kropat et al., 2005). Crr1 is also affected by Ni and Co, which act antagonistically to Cu in regulating the expression of Crr1 targets, suggesting metal-specific modulation. Although a similar mechanism in AtSPL7 has not been fully established, Cd has been shown to antagonize Cu by activating AtSPL7 targets even under Cu sufficiency (Gayomba et al., 2013). This supports the possibility that AtSPL7 may also act as a direct Cu sensor, potentially *via* Zn displacement in the SBP. Using the metal ion-binding site prediction and modelling server (MIB2, Lu et al., 2022), we identified four potential Zn, Cu, Ni, Co and Cd binding sites: His$^{475}$ Tyr$^{477}$, His$^{628}$Leu$^{630}$, $^{662}$SDIHRKH$^{668}$ and $^{644}$HCTCDCD$^{650}$ at the C-terminal region. There are also two CCC motifs (residues 564–566 and 683–685), in addition to scattered Cys and His residues. Whether these residues are involved in metal binding and Cu status sensing by AtSPL7 is yet to be determined.

### 4.2. Systemic signalling of Cu deficiency

Beyond local regulation, plants integrate whole-plant Cu status through systemic signalling mechanisms. AtSPL7 monitors Cu availability locally, especially near the vasculature (Araki et al., 2018). However, roots can also respond to shoot Cu demands via long-distance signalling, and Cu levels in the phloem contribute to shoot-to-root communication (Chia et al., 2023).

The strongest evidence for systemic Cu signalling comes from studies of AtOPT3, originally identified as an essential component of Fe homeostasis and signalling. AtOPT3 loads Fe into phloem companion cells; its disruption in *opt3* mutants leads to constitutive

Fe deficiency responses in roots and Fe accumulation in shoots due to a defect in Fe loading into phloem companion cells (Mendoza-Cózatl et al., 2014; Stacey et al., 2007; Zhai et al., 2014). Remarkably, this constitutive root Fe response can be repressed by Fe application to the shoot, indicating a phloem-mediated feedback signal (Chia et al., 2023).

We recently demonstrated that AtOPT3 also transports Cu ions when expressed in *Xenopus* oocytes and *S. cerevisiae* (Chia et al., 2023). The *opt3-3* mutant exhibits lower Cu in the phloem sap and impaired Cu recirculation from source to sink tissues (Chia et al., 2023). While the ability of AtOPT3 to transport Cu as well as Fe is not surprising, considering the recognized multispecificity of metal ion transporters, it was surprising to find, however, that the mutant also manifested the increased expression of Cu-deficiency marker genes in root tissues while overaccumulating Cu in the vascular tissues of the shoots (Chia et al., 2023). Importantly, these defects are rescued by Cu application. Furthermore, Cu feeding *via* the phloem in the shoot rescued the molecular symptoms of Cu deficiency in the root of the wild-type and the *opt3-3* mutant, suggesting the existence of long-distance shoot-to-root Cu signalling. This suggestion is further strengthened by results from reciprocal grafting experiments using wild-type and the *opt3-3* mutant (Chia et al., 2023). Thus, Cu availability in the shoot, mediated through phloem transport by AtOPT3, serves as a systemic signal to fine-tune root Cu uptake, ensuring balanced nutrient distribution across the plant. Interestingly, phloem-feeding with Cu in the shoot also rescued molecular symptoms of Fe deficiency in the root of the *opt3-3* mutant and decreased the transcript abundance of molecular markers of Fe deficiency in the root of wild-type. Likewise, phloem-feeding with Fe in the shoot downregulated the expression of both Fe- and Cu-deficiency marker genes in the root of the *opt3-3* mutant or wild-type, highlighting the complexity of the crosstalk between Cu and Fe in long-distance signalling. Together, these findings illustrate that Cu homeostasis in *A. thaliana* is maintained through a finely tuned network of local and systemic regulatory mechanisms. SPL7 emerges as a central integrator of Cu sensing and transcriptional responses, coordinating both Cu uptake and the economy/metal switch. In parallel, systemic signalling, mediated through phloem-localized transporters such as OPT3, ensures root Cu acquisition is aligned with whole-plant Cu status and is coordinated with Fe uptake and transport pathways.

## 5. Future perspectives

Although significant progress has been made in understanding Cu uptake strategies in non-grass species, the mechanisms underlying Cu uptake and long-distance transport in grasses remain incompletely defined. Multiple, potentially co-existing pathways for Cu acquisition, including direct uptake of $Cu^+$ and $Cu^{2+}$ and transport of Cu-ligand complexes, warrant further exploration. Furthermore, the identification of YSL transporters such as BdYSL3, which mediate Cu redistribution to reproductive tissues and influence grain size and protein content, underscores the need to investigate the role of Cu in shaping these essential agronomic traits. These studies would lay the groundwork for future exploration of the full suite of Cu transporters, ligands and regulatory networks that support Cu nutrition in cereal crops.

In addition to expanding our understanding of Cu uptake strategies across plant species, knowledge of internal Cu transport systems needs to be expanded as well. For example, the transporters responsible for Cu movement into and out of mitochondria have

yet to be identified. Cu sensing mechanisms, roles of Cu in local and systemic signalling and molecular basis of crosstalk with other mineral nutrients, represent additional promising avenues of investigation. Continued exploration of these pathways will be critical for understanding how plants optimize Cu allocation under fluctuating environmental conditions.

**Open peer review.** To view the open peer review materials for this article, please visit http://doi.org/10.1017/qpb.2025.10027.

## Acknowledgements

The authors would like to acknowledge and apologize to colleagues whose work was either not included due to a specific review focus or not cited due to space limitations.

**Competing interest.** The authors declare no competing interests.

**Data availability statement.** This review article does not rely on original data or resources.

**Funding statement.** The research in the O.K.V. lab is currently funded by U.S. National Science Foundation (US-NSF) Division of Integrative Organismal Systems (IOS) Award#: 2430791; U.S. Department of Agriculture National Institute of Food and Agriculture (USDA-NIFA) Award#: 2021-67013-33798 and Research Capacity Fund (HATCH) Award#: 2024-25-153, from the USDA-NIFA. T.O.Z.'s studies in the O.K.V. lab is funded by the USDA-NIFA Workforce Development Award#: 2021-67034-35124; J.C. is, in part, supported by the US-NSF Division of Biological Infrastructure (DBI) Award#: 2330043.

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
