## [Reviewer Report]

The authors have made a great effort in summarizing the latest advances in this hot topic; and also, in comparing the differences in the regulation of the Cu homeostasis in dicots and monocots.

I have not found any major mistakes. I would just suggest reviewing the manuscript looking for minor typos.

The figures are specially well selected and designed, with clear and understandable illustrations.

---

## [Reviewer Report]

The review summarizes research on copper homeostasis in Arabidopsis thaliana. Despite a few possible improvements, the review is thorough and interesting to the field. However, authors claim in the abstract that they cover Cu homeostasis in grasses, as well as iron and copper interactions. This is not entirely true for the current version. Important literature from rice is missing, such as papers describing OsSPL9 function, the ortholog of AtSPL7 (10.1111/nph.70074 and 10.1093/jxb/erae273); the possible existence of Cu economy in rice (10.1016/j.plaphy.2020.11.051); papers on transporters involved in Cu homeostasis (10.1016/j.jhazmat.2024.135245, 10.1038/ncomms12138, 10.1016/j.jhazmat.2021.128063), and others. Also, extensive literature on Fe and Cu interaction also exists, and was reviewed elsewhere, but not here. For example, IMA1 and CITF1 are linked to Fe/Cu crosstalk, and mechanistic insight is available for that (10.1111/nph.19439). Transcription factors involved in crosstalk have also been described (10.1093/jxb/erad439). I suggest authors either narrow the scope of the review, or add to these topics.

Other comments:

• Line 42: please make explicit the reason why the Irvine-Williams series explains that.

• Line 97-98: please make sure you check the recent paper by Rober et al (10.1371/journal.pgen.1011796), which shows more clearly a role of ZIP2 in Cu uptake.

• Line 107-109: please make sure you reference the literature on NA having a role in long distance internal transport by binding Fe(II) (maybe you can combine with the next paragraph). The mentioning of chloronerva seems detached, so maybe authors can give more context.

• Line 119-121: the phrasing is not clear. “loss of function phenotype” is not a phenotype – the mutant is a loss of function one (i.e., knockout mutant), but you need to tell which specific phenotype you are referring to (e.g., constitutive chlorosis? High/low Fe or Cu accumulation? Etc). Otherwise “intermediate phenotype” has no meaning (as it would imply “intermediate loss of function”). Please correct.

• Line 168-171: this small paragraph also seems out of context. Please detail a bit more about the chaperones and metallothioneins roles in Cu transport.

• Line 183: please correct brachypodium to italics. Please check other instances as well.

• Line 196-197: please make sure you check the recent paper by Rober et al (10.1371/journal.pgen.1011796), which shows more clearly a role of ZIP2 in Cu uptake.

• Line 279: there is one OsHMA6 described (http://dx.doi.org/10.1016/j.rsci.2020.01.005). Is it an ortholog of AtHMA6? If yes, please discuss that in this section.

• Line 308: you did not mention OsHMA4, a known vacuolar Cu transporter well characterized in rice (10.1038/ncomms12138 and 10.1016/j.jhazmat.2021.128063).

• Line 335: “the essentiality” should be “and the essentiality” maybe?

• Line 362: “ensure Cu redistribution among essential cellular functions” is unclear. Please rephrase to make clear that this is the Cu economy mode, and that these functions are non-essential, for example.

---

## [Editor Report]

Dear Olena,

your manuscript has now been seen by two reviewers. I apologize for the delay, but reviewers were busy with conferences. Both reviewers are very positive and think that the MS makes a substantial contribution to the field. One reviewer mentioned a few points that might help to polish the MS in a minor revision. Please have a look at their comments and see how you can implement their suggestions. Thank you very much for your valuable contribution to the Research Topic “Quantitative approaches to cellular aspects of plant ion homeostasis”.

Best wishes, Ingo

---

## [Editor Report]

Dear Olena,

thank you for the careful revision of your manuscript and thanks again for your valuable contribution to the Research Topic “Quantitative approaches to cellular aspects of plant ion homeostasis”. It is highly appreciated.

Best regards, Ingo